# Detection of a Novel, and Likely Ancestral, Tn*916*-Like Element from a Human Saliva Metagenomic Library

**DOI:** 10.3390/genes11050548

**Published:** 2020-05-14

**Authors:** Liam J. Reynolds, Muna F. Anjum, Adam P. Roberts

**Affiliations:** 1UCD School of Biomolecular and Biomedical Science, University College Dublin, 4 Dublin, Ireland; liam.reynolds@ucd.ie; 2Department of Bacteriology, Animal and Plant Health Agency, Addlestone KT15 3NB, UK; muna.anjum@apha.gov.uk; 3Department of Tropical Disease Biology, Liverpool School of Tropical Medicine, Pembroke Place, Liverpool L3 5QA, UK

**Keywords:** saliva metagenome, Tn*916*, antibiotic resistance, *orf9*, tetracycline resistance, integrative conjugative element

## Abstract

Tn*916* is a conjugative transposon (CTn) and the first reported and most well characterised of the Tn*916*/Tn*1545* family of CTns. Tn*916*-like elements have a characteristic modular structure and different members of this family have been identified based on similarities and variations in these modules. In addition to carrying genes encoding proteins required for their conjugation, Tn*916*-like elements also carry accessory, antimicrobial resistance genes; most commonly the tetracycline resistance gene, *tet*(M). Our study aimed to identify and characterise tetracycline resistance genes from the human saliva metagenome using a functional metagenomic approach. We identified a tetracycline-resistant clone, TT31, the sequencing of which revealed it to encode both *tet*(M) and *tet*(L). Comparison of the TT31 sequence with the accessory, regulation, and recombination modules of other Tn*916*-like elements indicated that a partial Tn*916*-like element encoding a truncated *orf9* was cloned in TT31. Analysis indicated that a previous insertion within the truncated *orf9* created the full length *orf9* found in most Tn*916*-like transposons; demonstrating that *orf9* is, in fact, the result of a gene fusion event. Thus, we hypothesise that the Tn*916*-like element cloned in TT31 likely represents an ancestral Tn*916*.

## 1. Introduction

Tn*916* is the archetype of the Tn*916*/Tn*1545* family of broad host range conjugative transposons or integrative and conjugative elements (CTns and ICEs respectively), members of which have been described in over 35 genera of bacteria [1,2,3]. CTns, or ICEs, are mobile genetic elements that can mediate their own transfer from one cell to another via direct cell to cell contact and a type IV secretion system dependent conjugation process. They carry all the necessary genes that encode for the proteins required for their excision from (and integration into) the host genome, their conjugal transfer to a recipient cell and, as recently proven, their independent replication as an excised circular molecule [4,5]. Tn*916*-like elements have modular structures; they contain characteristic conjugation, regulation modules and usually a tyrosine integrase comprising a recombination module plus more variable accessory genes that almost always contain antimicrobial resistance genes; primarily *tet*(M). The gene *tet*(M) encodes a ribosomal protection protein conferring resistance to tetracyclines although genes conferring resistance to antibiotics other than tetracycline such as macrolides (*erm*(B); accession no. AM410044), biocides (*qrg*; accession no. HQ663849) and heavy metals (*merA*; accession no. EU239355) have also been identified on Tn*916*-like elements [1,3,6,7]. Although the genes present in these modules can differ between different members of this family their modular structure can still be used as a basis to identify them [3,8].

The array of accessory genes identified on Tn*916*-like elements is evidence of the continuing gene acquisition events that play an important role in the evolution of this CTn family. For example, *erm*(B) is associated with IS*1216* insertion sequences (ISs) on Tn*1116* [9]. In fact, IS*1216* transposases have shaped the structure of other Tn*916*-like elements including Tn*6087* and Tn*6079* by introducing *qrg* and *erm*(T) antimicrobial resistance genes respectively onto a Tn*916* backbone [1,10]. Exchanges between Tn*916*-like elements and introns, plasmids and other transposons have also been described [11,12,13]. For example; both Tn*5397* and Tn*6000* carry a group II intron inserted into *orf14* and so this insertion may be a point of divergence between these two CTn’s and the Tn*916* progenitor which does not contain a group II intron [14,15]. Genetic reorganization events have not been limited to just the acquisition of accessory genes. Genes involved in the regulation of conjugation vary between Tn*916*-like elements. For example, CTn*1* from *Clostridioides difficile* and Tn*5386* do not encode *orf9*, *orf10*, *orf5* and *orf6*. This further demonstrates that gene acquisition and deletion play a role in the Tn*916*-like element evolution. It also suggests that different Tn*916*-like elements employ different regulatory pathways despite the central role as a transcriptional repressor being proposed for *orf9* [16,17].

Evolution of other CTn families via gene acquisition, deletion and recombination have also been reported. Wozniak et al. demonstrated that there is a minimum gene structure required for SXT/R391-type conjugal transfer and that acquisition of genes within five “hotspots” of this core structure shaped the evolution and diversity of this CTn family [18]. Similarly, CTn’s of the ICE*St1*/ICE*St3* family have a defined core structure and a variable region where gene acquisition has been described. ICE*St1* and ICE*St3* differ in that the former encodes a bacteriophage resistance system in its variable region and the latter encodes putative methyltransferase genes [19,20]. 

Multiple studies have reported the identification of antibiotic-resistant bacteria and resistance genes from the human oral cavity. Indeed, the association of antibiotic resistance genes with Tn*916*-like elements from oral bacteria, in particular *Streptococcus* spp., has been described [21,22]. As estimates indicate that only two-thirds of the bacteria from this environment have been cultured we created and functionally screened, a human saliva metagenomic library to identify antibiotic resistance genes and associated mobile genetic elements [23]. We have previously reported on the novel efflux genes *tet*AB(60) encoding resistance to tetracycline and tigecycline in *Escherichia coli* and multiple genes encoding resistance to biocides from this metagenomic library [24,25].

In this manuscript, we report on a tetracycline-resistant clone identified in this metagenomic library designated TT31. It contains both *tet*(M) and *tet*(L), encoding a tetracycline efflux gene. Both genes are part of a novel Tn*916*-like element. Analysis of the clone indicates that the DNA likely originated from a host that shares sequence homology to *Gemella* sp. that also harbors what we propose to be an ancestral Tn*916*-like element.

## 2. Materials and Methods

### 2.1. Bacteria Strains and Culture Conditions

*E. coli* EPI300 clones were cultured in Luria-Bertani broth (LB; Sigma-Aldrich^®^, Gillingham, UK) and LB agar (Life Technologies^TM^, Renfrew, UK) at 37 °C with shaking at 200 rpm for liquid cultures. Antibiotic selection was achieved by supplementing media with chloramphenicol (12.5 μg/mL; Sigma-Aldrich^®^) and tetracycline (5 μg/mL; Sigma-Aldrich^®^). 

### 2.2. Creation of a Functional Metagenomic Library

Metagenomic DNA was extracted from the pooled saliva of 11 individuals (in 1.5 mL aliquots) who had not received antibiotics within the previous 3 months. The UCL Research Ethics Committee granted ethical approval to collect human saliva from volunteers (Project ID Number 5017/001) and all participants gave written consent before participating. The human saliva metagenomic library was created using the pCC1BAC vector and the tetracycline sensitive (minimum inhibitory concentration of 2 µg/mL) strain TransforMax EPI300 Electrocompetent *E. coli* as previously described [24,25].

### 2.3. Screening of the Functional Metagenomic Library

From the human saliva functional metagenomic library, 27,000 clones were screened for tetracycline resistance by culturing the clones on LA supplemented with chloramphenicol (12.5 μg/mL) and tetracycline (5 μg/mL) at 37 °C for 16 h [24]. A tetracycline-resistant clone, TT31, was selected for further characterisation.

### 2.4. Plasmid Extraction

In 20 mL tubes, 1 mL of a 16 h culture of TT31 in LB supplemented with chloramphenicol (12.5 μg/mL) and tetracycline (5 μg/mL) was subcultured into 9 mL fresh LB supplemented with chloramphenicol (12.5 μg/mL), tetracycline (5 μg/mL) and 10,000X CopyControl™ Induction solution (10 µL in 10 mL). This subculture was then incubated horizontally (by taping the tube, on its side, to the base of the incubator) for 4 h at 37 °C with shaking at 200 rpm. The pCC1BAC vector was extracted from this TT31 clone subculture using the QIAprep Spin Miniprep Kit (QIAGEN, Manchester, UK) according to the manufacturer’s guidelines.

### 2.5. Sequencing and Analysis

The pCC1BAC based plasmid extracted from TT31 was sequenced using primer extension Sanger sequencing by Beckman Coulter Genomics Inc using the primers detailed in Appendix A. Assembly of the TT31 contig was conducted using SeqMan Pro (Lasergene software, DNASTAR, Madison, WI, USA). The TT31 sequence was submitted to GenBank (accession: MF344584) and is included in Appendix A. The sequences were analysed using the tools from the National Centre for Biotechnology Information (NCBI). Additional sequence data of interest were obtained from the NCBI database and alignments were made using the ClustalΩ software at http://www.ebi.ac.uk/Tools/msa/clustalo/.

## 3. Results

### 3.1. Sequence Analysis and Gene Annotation of TT31 Insert

Sequencing of the pCC1BAC::vector insert extracted from TT31 revealed it to be 14,171 bp in size. BLASTN analysis of the sequence showed that from 1 bp to 7226 bp (the left side of Figure 1a) had 98% nucleotide identity to *Streptococcus pyogenes* strain emm65 (99% cover; accession: CP035433). The left side of TT31 encoded nine ORFS including an excisionase (*xisTn*), a tyrosine integrase (*intTn*), *tet*(M) as well as a variant of *orf9*, *orf5*, *orf8*, *orf7* and *orf10* all of which are found on Tn*916*-like elements. An ORF exhibiting 99% nucleotide identity to *tet*(L) from several *Enterococcus* and *Streptococcus* species was also identified (Figure 1a,b).

TT31, from 7214bp to 14,171 bp, (the right side, Figure 1a), had 95% nucleotide identity to *Gemella haemolysans* NCTC 10459 (accession:LR134484.1). This region of TT31 encoded *yheH* and *yheI* (multidrug ABC transporter subunits), a NAD-dependent deacetylase and a SecA translocase; all of which have homology to proteins from *G. haemolysans* NCTC 10459 (>98%), (Figure 1a). 

As *Hin*dIII had been used to digest the oral metagenomic DNA for library construction we ascertained the position of any internal *Hin*dIII site to rule out the possibility of concatemer formation during cloning. No *Hin*dIII site was present at the boundary of the left side (exhibiting homology to *S. pyogenes* strain emm65) and the right side (exhibiting homology to *G. haemolysans* NCTC 10459) of TT31, indicating that the insert was not a concatemer and that it wasn’t a result of our cloning protocol.

### 3.2. Nucleotide Sequence Homology of the TT31 Insert with Tn916

BLASTN analysis revealed Tn*916* from *Enterococcus faecalis* (accession: U09422) to have 74% coverage and 97% nucleotide identity to the left 7226 bp of TT31 (Figure 1b) [8]. An alignment between 1 bp and 7226 bp of TT31 and the 5844 bp regulatory, recombination and accessory [*tet*(M)] modules of *E. faecalis* Tn*916* showed that the first 1874 bp of TT31 aligned with the *E. faecalis* Tn*916 tet*(M) (12,185 bp–14,058 bp). Between 3759 bp to 7216 bp, TT31 had 97% nucleotide identity to *E. faecalis* Tn*916* (14,571 bp–18,032 bp) (Figure 1b). This region of TT31 encoded *orf*10, *orf*7, *orf*8, *orf*5, the *xisTn* and *intTn* genes as well as the 5′-part of *orf9*, that are characteristic of Tn*916* [22,26]. Within TT31, no alignment was observed between 1875 bp–3759 bp and Tn*916*, this region contained *tet*(L) which is not encoded by Tn*916*; this explains the 74% coverage observed for this alignment.

### 3.3. TT31 Encodes a Truncated orf9

Interestingly, TT31 *orf9*, which we have designated *orf9t*, was truncated compared to that found on Tn*916*. The Tn*916 orf9* is an open reading frame of 354 nucleotides whereas *orf9t* (270 nucleotides) was 84 nucleotides shorter at the 3′ - end. The first 171 nucleotides of the Tn*916 orf9* showed >99% nucleotide identity with TT31 *orf9t*; the nucleotides following this point shared no homology (Figure 1c). The lack of homology between the 3′-ends of *orf9t* and Tn*916 orf9* was further demonstrated by aligning their putative amino acid sequences. Orf9t and Tn*916* Orf9 showed no amino acid homology after amino acid 57, which agreed with the nucleotide alignment (Figure 1d). Additionally, 14,058 bp–14,570 bp of Tn*916*, which encoded *orf6*, did not align with TT31 (Figure 1b). BLAST analysis of the *orf9t* sequence revealed it to be entirely present in the regulatory region of Tn*916*-like elements in *S. pyogenes* strain emm65, *S. pyogenes* NCTC 13737 (LS483425.1) and *Streptococcus agalactiae* strain 874,391 (CP022537.1).

### 3.4. A Plasmid Insertion Occurred within orf9t in Lactobacillus Johnsonii

Analysis of the first 7226 bp of TT31 revealed it to have 60% coverage and 97% nucleotide identity to a 12,313 bp section of the *Lactobacillus johnsonii* strain BS15 genome that encodes *tet*(L) (accession: CP016400). An alignment of this 7226 bp region of TT31 with the 12,313 bp region of the *L. johnsonii* strain BS15 genome showed that there is an insertion in the *L. johnsonii* sequence relative to TT31 which explains why there is only 60% coverage between the two sequences. This insertion is in *orf9t*, between nucleotides 3758 bp and 3759 bp as these nucleotides aligned to positions 41,612 bp and 46,612 bp respectively on the *L. johnsonii* genome (Figure 2). Four ORFs were found to be encoded in this intervening 4999 bp section of DNA. These ORFs encoded *pre/mob* and *rep* genes, an IS*110* family transposase and an *orf*6 homolog. The 183 bp of the 3′—end of *orf9* that is present on Tn*916* but not in TT31 was also contained in this insert (Figure 2). This 3′—end of *orf9* was in frame with the start codon containing 5′—end of *orf9t* such that *L. johnsonii* strain BS15, like Tn*916*, encoded the full length fusion *orf9* (Figure 2). The 99 bp 3′—end of *orf9t* was retained by the *L. johnsonii* genome, outside of the insertion (Figure 2). 

### 3.5. tet(L) and the 3′-orf9t Sequence Are Present in the TT31 Tn916-Like Element and Tn6079

Tn*6079* (accession: GU951538) is a CTn encoded by *Streptococcus gallolyticus* that was first identified from a functional metagenomic screen of the infant gut [10]. As Tn*6079* encoded *tet*(L) and had a similar genetic organisation to TT31, the left 7226 bp of TT31 was aligned with the 11,701 bp accessory, regulation and recombination modules of Tn*6079*. This revealed a 99% nucleotide identity (82% cover) between the two sequences (Figure 3). Alignment of TT31 (between 1 bp and 3758 bp) and Tn*6079* (between 17,176 bp and 20,917 bp) represents homology between the *tet*(M) and *tet*(L) encoding sections of the sequences. It also confirms that the 99 bp 3′-end of *orf9t* is present on Tn*6079*. The *orf5*, *xisTn* and *intTn* encoding regions of TT31 (between 4993 bp and 7226 bp) and Tn*6079* (between 26,639 bp and 28,876 bp) also aligned (97% nucleotide identity) (Figure 3). The *orf8*, *orf7*, *orf10* and the 5′—end of *orf9t* encoding region of TT31 (3759 bp–4993 bp) did not align with Tn*6079*. Additionally, the accessory Tn*6079* genes encoding a plasmid recombination protein (*pre/mob*), a plasmid replication protein (*rep*) and *erm*(T) that is flanked by IS*1216* sequences (20,918 bp–26,639 bp) did not align with TT31, (Figure 3). The *pre/mob* and *rep* genes of Tn*6079* had >99% nucleotide identity to those identified in the *L. johnsonii* strain BS15 genome insertion. 

## 4. Discussion

### 4.1. A Partial Tn916-Like Element Was Cloned in TT31

Analysis of the TT31 insert showed the left 7226 bp had homology to a Tn*916*-like element from *S. pyogenes* strain emm65 and the right side of the TT31 insert had homology to *G. haemolysans* NCTC 10459. The lack of *Hin*dIII restriction site between the left and right sides of the cloned insert indicated that it was not a concatemer. Therefore, the cloned insert contained a partial Tn*916*-like element from a host sharing homology with a *Gemella* sp. Tetracycline resistance mediated by *tet*(M) has been described in *Gemella* spp. before; evidence for its association with the Tn*916* integrase *(intTn)* in this genus was reported when Zolezzi et al. [27] used PCR to identify these markers in nasopharyngeal *Gemella* spp. isolates, however, beyond this study little work has been conducted regarding Tn*916*-like elements in this genus [27,28,29]. To the best our knowledge *tet*(L) encoding *Gemella* spp. have not been described in the literature. However, it should be noted that sequence analysis of the small section of DNA cloned in TT31 is not sufficient to identify the exact genera from which the cloned DNA originated and so it is possible the DNA originated from a non-*Gemella* sp.

### 4.2. The Impact of tet(L) Expression on a Tn916-Like Element

Transcription of *tet*(M) and surrounding genes are proposed to play an important role in the regulation of Tn*916* excision and integration. In the absence of tetracycline, *tet*(M) transcription is hypothesised to be prevented by stem-loop structures in the upstream, *orf12* RNA that halts the progress of the RNA polymerase through *tet*(M) and beyond [30]. In the presence of subinhibitory concentrations of tetracycline most ribosomes are inhibited and so charged tRNA molecules accumulate within the cell. Under these conditions, some ribosomes are thought to be protected by basal levels of Tet(M). These protected ribosomes can rapidly transcribe the *orf12* transcript as charged tRNA molecules are more readily available and this prevents stem-loop structures forming, resulting in increased *tet*(M) transcription as the RNA polymerase is no longer halted. Increased transcription through *tet*(M) is proposed to produce antisense *orf9* RNA. Orf9 has putative DNA binding domains and is believed to repress transcription of *orf7;* an up-regulator of *xisTn* and *intTn* expression. As such the production of antisense *orf9* RNA de-represses *orf7* resulting in excision of the Tn*916* element from the host genome, circularisation of the excised element and increased transcription, across the ligated ends of the element, of the conjugation related genes on the other end of the CTn [31,32]. 

In the context of the TT31 Tn*916*-like element, transcription through *tet*(M) as described above would also result in increased transcription of *tet*(L) which is immediately downstream of it. Tet(L) is an antiporter of the major facilitator superfamily that pumps tetracycline out of the cell using the proton motive force generated by the concomitant uptake of a proton [33,34]. Thus, increased expression of Tet(L) could act to reduce tetracycline levels in the cell below the threshold concentration required to initiate excision of the CTn as described above. Tet(L) could, therefore, act as a second gauge that controls the excision of the CTn under unfavorable conditions, such as low pH, that oral bacteria frequently encounter [35]. Such a scenario would offer some explanation as to why *tet*(L) has been identified downstream of *tet*(M) in only a few Tn*916*-like elements as the excision of such elements could be more tightly regulated and therefore, they may have disseminated less. The loss of *tet*(L) could have loosened this control allowing CTn’s expressing only *tet*(M) to excise and disseminate more readily. 

The role of antisense *orf9* production and its role in regulating the excision of *tet*(M) and *tet*(L) encoding Tn*916*-like elements requires further study. To date Tn*6079* is the only *tet*(M) and *tet*(L) encoding Tn916 element that has been described in detail. Tn*6079* does not encode *orf9*, however, and so regulation of its excision is by some alternative pathway [10]. 

The maintenance of *tet*(L) in addition to *tet*(M) by some Tn*916*-like elements may be a result of *tet*(L) expression offering some additional benefit to the host cell. Indeed, Tet(L) has been shown to accept Na^+^ and K^+^ as substrates and may act as an important regulator of these ions under conditions of stress [36]. For example, *B. subtilis* strains in which *tet*(L) has been deleted are more sensitive to high pH than their wild type *tet*(L) expressing counterparts [37].

### 4.3. The TT31 Insert Contained a Partial ‘Ancestral’ Tn916-Like Element

The results of our analysis indicated that TT31 contained a partial and likely “ancestral” Tn*916* element and that a series of hypothetical events of acquisition and subsequent loss of genes by this ‘ancestral’ transposon culminated in the formation of Tn*916*. In this evolutionary scenario, the Tn*916*-like element encoded by the *L. johnsonii* strain BS15 genome is representative of an ‘intermediate’ between Tn*916* and the TT31 Tn*916*-like element. The series of events may have taken place as follows. A *pre/mob*, *rep*, transposase and *orf6* encoding plasmid may have inserted into *orf9t* of the TT31 Tn*916*-like element. This plasmid also contained the 3′—183 bp of the full length *orf9* gene, from Tn*916*, and its insertion into the TT31 Tn*916*-like element resulted in the creation of the Tn*916*-like element identified in *L. johnsonnii* strain BS15. *Staphylococcus aureus* strain ADB2006 (MH8232.16) carries a pathogenicity island that encodes sequences with 75% and 85% nucleotide identity to Tn*916 orf6* and the 3′—183 bp of full length *orf9,* respectively. The *orf6* and the 3′—183 bp of the full length *orf9* may have been acquired by the *pre/mob*, *rep*, transposase encoding plasmid prior to insertion in the TT31 Tn*916*-like element (Figure 4). The existence of these sequences outside of a Tn*916*-like element supports our hypothesis that the full length Tn*916 orf9* is the result of a gene fusion that resulted from an insertion event into *orf9t*.

This plasmid insertion was followed by an imprecise excision or recombination event, potentially mediated by the IS*110* transposase or the *pre*/*mob* associated recombinase, whereby sites outside of the plasmid were recognised resulting in the excision of a region of the CTn containing *tet*(L) and the *pre/mob*, *rep* and transposase genes but leaving *orf6* and the full length *orf9* fusion behind, thus creating Tn*916* (dotted box in Figure 4). Observations of imperfect excision events have been described previously. For example, Laverde Gomez et al. identified that fragments of the pathogenicity island of *E. faecalis* strain UW3114 remained in the chromosome following its excision. The authors concluded this imprecise excision occurred due to the erroneous recognition of sequences flanking the element that exhibited homology to internal sequences required for excision [38]. 

### 4.4. Tn6079 Descends from the TT31 Tn916-Like Element

Similarly, a relationship between Tn*6079* and TT31 can be hypothesised where TT31 is an ancestral Tn*916*-like element to Tn*6079* with the *L. johnsonii* Tn*916*-like element as an ‘intermediate’. As previously described for the hypothetical evolution of Tn*916*, insertion of a plasmid containing *pre /mob*, *rep*, an IS*110* transposase and *orf6* genes and the 3′—183 bp of the full length *orf9* inserts into *orf9t* of TT31, creating the ‘intermediate’ *L. johnsonii* Tn*916*-like element. Deletion of *orf10*, *orf7*, *orf8*, *orf6* and *orf9* then occurs following a recombination event between two regions of this CTn. This is similar to the event hypothesised by Croucher et al. to have resulted in the deletion of *aphA*-3 in a *S. pneumoniae* Tn*916*-like element [39]. Following this recombination event, an IS*1216* element carrying *erm*(T) inserts upstream of *orf5* giving rise to Tn*6079* (Figure 5). It is also plausible that a lone IS*1216* element inserted upstream of *orf5* followed by the insertion of a second IS*1216* type transposable unit containing *erm*(T). IS elements may flank accessory genes, including antibiotic resistance genes and can mediate their transposition; indeed IS*1216* has been implicated in the dissemination of glycopeptide resistance genes and has been found flanking the *cfr* resistance gene in the pEF-01 plasmids isolated from *E. faecalis* [40,41].

## 5. Conclusions

In summary, we have identified a novel Tn*916*-like element from the human saliva metagenome using a functional metagenomic approach. As this Tn*916*-like element encoded an *orf9* variant we have hypothesised that it is from an ancestral CTn to Tn*916*. These hypotheses point to a Tn*916*-like element with a gene organisation as described in TT31 as a predecessor of Tn*916* CTns and that *L. johnsonii* strain BS15 encodes a representative of an intermediate Tn*916*-like element. The presence of the 3′-end remnant of *orf9t* in both *L. johnsonii* strain BS15 and Tn*6079* supports the directionality of the events that we have described in this work. This directionality is further supported by the identification of homologs of *orf6* and the 3′-end of the full length *orf9* in a *Staphylococcal* pathogenicity island, which indicates that *orf9* is a fusion gene. This work highlights the importance of *orf9* as a point of evolutionary divergence in the Tn*916*/Tn*1545* family of CTns. 

## Figures and Tables

**Figure 1 genes-11-00548-f001:**
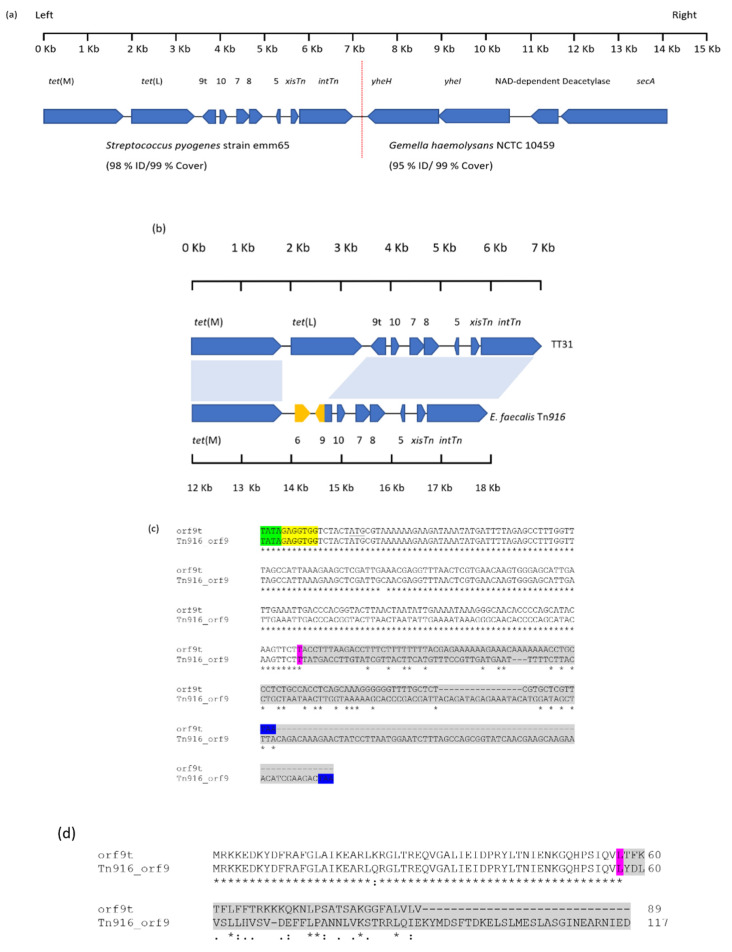
Alignment of TT31 and Tn*916*. (**a**) A schematic of the ORFs present on the TT31 Clone. The vertical dashed red line highlights the border of the sequences with nucleotide homology to *Streptococcus pyogenes* strain emm65 on the left and on the right *Gemella haemolysans* NCTC 10459. *orf9t*, *orf10*, *orf7*, *orf8*, and *orf5* are shown as 9t, 10, 7, 8 and 5 respectively. (**b**) TT31 and Tn*916* with aligned regions indicated by blue shaded areas. Regions of Tn*916* that are not present in TT31 are highlighted in yellow. *tet*(M) is found in TT31 and Tn*916* as are the Tn*916* regulation and recombination modules. *tet*(L), however, is not present on Tn*916* and *orf6* is not encoded by TT31. Furthermore, *orf9* is truncated in TT31 (designated orf9t) compared to Tn*916 orf9*. (**c**) A Clustal Ω alignment between *orf9t* and Tn*916 orf9*. The start codons are underlined and stop codons are highlighted blue. The predicted ‘TATA box’ and Shine-Dalgarno sequences are highlighted green and yellow respectively for both. The purple highlighted nucleotides identify the point of divergence between *orf9* and *orf9t*. The grey shaded area represents the 99 nucleotides and 183 nucleotides of the 3′-end of *orf9t* and *orf9* respectively that do not share homology. The stop codon for both are highlighted blue. (**d**) A Clustal Ω alignment between the amino acid sequences of Orf9t and Tn*916* Orf9. The purple highlighted amino acids represent the point of divergence of homology between the two proteins and represents the same point of divergence observed for the nucleotide alignment. The grey shaded amino acids are those that share no homology at the N-terminal end of both proteins.

**Figure 2 genes-11-00548-f002:**
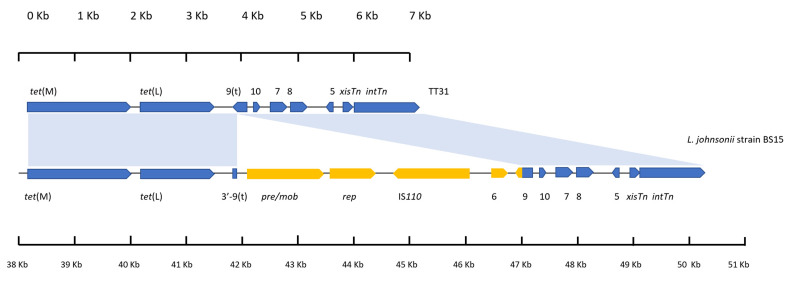
An alignment of the left 7226 bp of TT31 and a 12,313 bp region of the *L. johnsonii* strain BS15 genome. Blue shaded areas represent areas of alignment between the two sequences. Individual scales accompany each gene schematic. *tet(*M) and *tet*(L) are encoded by both as are *orf10, orf7, orf8, orf5*, *xisTn* and *intTn*. *orf9t* is encoded by TT31. An insertion is present in the *orf9t* of *L. johnsonii* strain BS15 relative to TT31. *pre/mob*, *rep*, IS*110* transposase and *orf6* genes are present on this insert; highlighted in yellow. Also present on this insert is the 183 bp 3′—end of the full length *orf9* (highlighted in yellow) in frame with the start codon of the 5′ end of *orf9* that is found in all the sequences (highlighted in blue); thus the full length *orf9* is encoded by *L. johnsonii* strain BS15. The *L. johnsonii* genome still maintains the 3′—end of *orf9t* (highlighted blue).

**Figure 3 genes-11-00548-f003:**
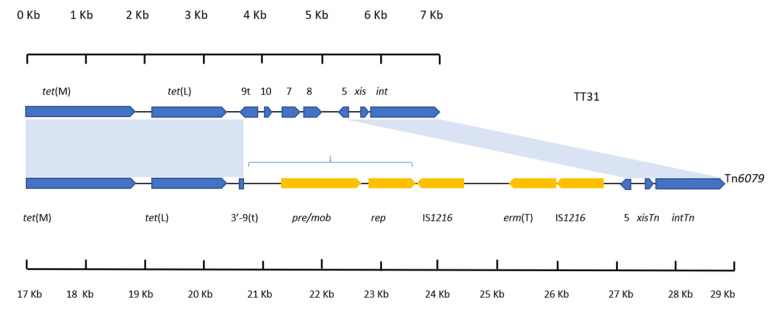
An alignment between the left 7226 bp of TT31 and the right 11,701 bp of Tn*6079*. Regions of Tn*6079* and TT31 that aligned are indicated by the blue shaded areas. Individual scales accompany each gene schematic. *tet*(M) and *tet*(L) are present in both as are *orf5*, *xisTn* and *intTn*. *orf9, orf10, orf7* and *orf8* are not present on Tn*6079*. The *pre*/*mob*, *rep* and *erm*(T) genes (the latter of which is flanked by IS*1216* sequences) that are encoded by Tn*6079* are not present on TT31 and are highlighted in yellow. The region of Tn*6079* highlighted by blue brackets is present in the *L. johnsonii* BS15 genome also.

**Figure 4 genes-11-00548-f004:**
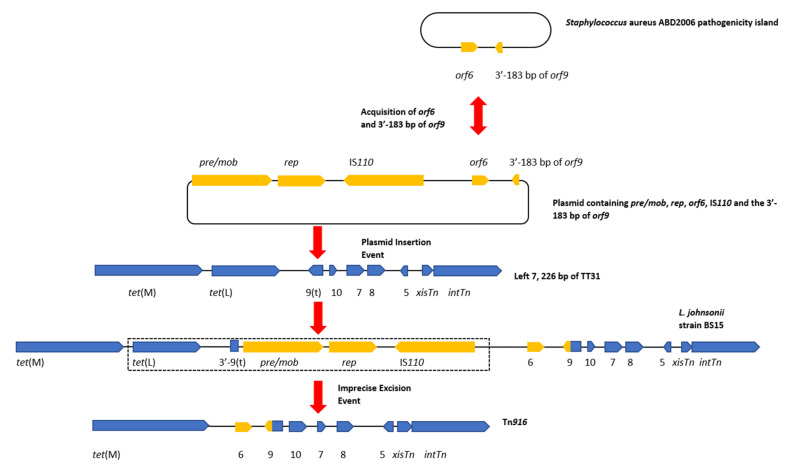
A diagram of the hypothetical events that led to the creation of Tn*916*. This hypothetical event assumes that the left 7226 bp of TT31 are a part of a larger Tn*916*-like element. Gene diagrams are not to scale. A plasmid containing *pre/mob* and *rep* genes and an IS*110* transposase obtains *orf6* and the 3′—end of *orf9* (yellow) from the *Staphylococcus aureus* ABD2006 pathogenicity island. A double-headed arrow between the *Staphylococcus aureus* ABD2006 pathogenicity island and the plasmid indicates that *orf6* and the 3′-183 bp of *orf9* may have transferred in either direction. Both scenarios demonstrate the mobility of this region of DNA. Insertion of this plasmid then occurs within *orf9t* of the TT31 Tn*916*-like element which creates the *L. johnsonii* strain BS15 encoded ‘intermediate’ Tn*916*-like element. This plasmid insertion disrupts the *orf9t* reading frame resulting in the creation of the larger fusion *orf9* (yellow and blue) although the 3′—end of *orf9t* is still present in *L. johnsonii* strain BS15. An imprecise excision event then occurs which results in the loss of the *pre/mob*, *rep*, IS*110* transposase and *tet*(L) genes as well as the 3′—end of *orf9t* (indicated by a dotted box), creating Tn*916*.

**Figure 5 genes-11-00548-f005:**
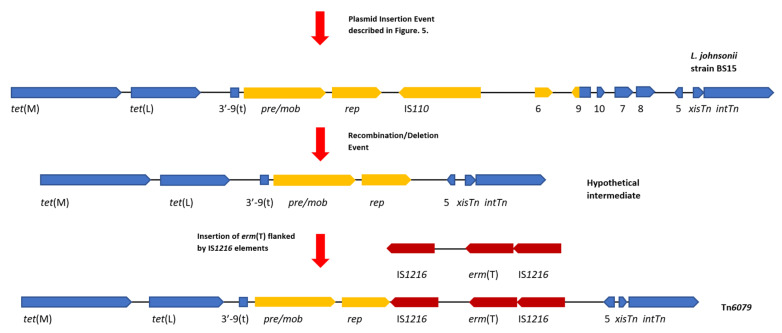
A diagram of the hypothetical events that created Tn*6079*. This hypothetical event assumes that the left 7226 bp of TT31 are part of a larger Tn*916*-like element. Gene diagrams are not to scale. Following on from the insertion event that created the *L. johnsonii* strain BS15 ‘intermediate’ Tn*916*-like element described in Figure 4, a deletion/recombination event results in the loss of *orf6*, *orf9*, *orf10*, *orf7* and *orf8*, forming a hypothetical intermediate CTn. Insertion of *erm*(T) flanked by IS*1216* elements (dark red) follows this deletion event creating Tn*6079*. For continuity with Figure 4, *tet*(M), *tet*(L), *orf5*, *xisTn* and *intTn* and the 3′—end of *orf9*(t) are coloured blue and the plasmid associated genes are yellow.

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
