# Peer review of "Detection of a Novel, and Likely Ancestral, Tn916-Like Element from a Human Saliva Metagenomic Library"

_genes, 2020, doi:10.3390/genes11050548_

Round 1

Reviewer 1 Report

Reynolds et al. report the detection and bioinformatic analysis of a Tn916-like element. They make the case that the element they have found represents an ancestral genomic configuration relative to known Tn916s, and present a scenario for its evolution.

While this study has a relatively narrow focus, I think the provided scenario is valuable was interesting to me.

Larger points

- The authors state with great certainty that this came from a Gemella sp. host. I may be missing something here. But why is it not possible that it rather came from something more closely related to Streptococcus pyrogenes? Indeed, can we be certain that it didn't come from something altogether different? If the case really is this strong, please make it much more clearly.

- In presenting the scenario I think it's important to stress why it is more likely that events went in this direction rather than the other way around. (ie, why not Tn916 ancestral to TT31). I think one answer to this is the fact that the BS15 strain has an extra bit of the 3' end of orf9. Authors should clarify this.

- The point of figure 1c is to show that the right part doesn't align well. To be extra careful, it would be best to do a protein alignment, and then back align to get the nucleotides. Likely will give the same result, but then we'll be really confident there's no homology lurking there.

Smaller points

- The introduction focuses exclusively on Tn916. I think it would benefit from a little discussion on what is known about the evolution of other CTns. Are there other cases where people have put together scenarios like the one being proposed here?

- Fig 1 b suggestions. Color the E. faecalis Tn916 element differently, and then make the axis below (giving distances in Tn916) have a similar color so that it matches visually.
Also, there is a bit of a gap between the drawing of the genes and this axis, which makes them not group together visually as well as they could.
Make the font of TT31 and E. faecalis Tn916 bigger. This should be prominent.

- Fig 1 c. Since Orf9t was on the top in 1b, it makes sense to put it on the top in 1c.

Author Response

Reynolds et al. report the detection and bioinformatic analysis of a Tn916-like element. They make the case that the element they have found represents an ancestral genomic configuration relative to known Tn916s, and present a scenario for its evolution.

While this study has a relatively narrow focus, I think the provided scenario is valuable was interesting to me.

Larger points

- The authors state with great certainty that this came from a Gemella sp. host. I may be missing something here. But why is it not possible that it rather came from something more closely related to Streptococcus pyrogenes? Indeed, can we be certain that it didn't come from something altogether different? If the case really is this strong, please make it much more clearly.

Response

The TT31 insert sequence shares nucleotide homology to Streptococcus pyogenes, however, this sequence homology resides within a Tn916-like mobile genetic element. When identifying the source of cloned DNA from a functional metagenomic screen when a mobile genetic element is present, it is accepted practice within metagenomic studies to use the sequence of flanking chromosomal DNA (in this case highest homology to Gemella sp.) rather than the genetic element itself due to its mobility. The flanking DNA that shares homology with Gemella sp. encodes chromosomal genes and the authors believe this better shows what the source of the cloned DNA is.

However, the authors also recognise that the DNA may not have originated from a Gemella sp. and have made amendments to the manuscript on lines 77, 152, 231 and 236 have been included to clarify that the sequence has the highest homology to Gemella sp. but that it is possible that the DNA originated from a bacterium of a different genera.

- In presenting the scenario I think it's important to stress why it is more likely that events went in this direction rather than the other way around. (ie, why not Tn916 ancestral to TT31). I think one answer to this is the fact that the BS15 strain has an extra bit of the 3' end of orf9. Authors should clarify this.

Response

The reviewer is correct in highlighting that the directionality of the authors proposed series of events is supported by the presence of the 3' end of orf9t in the BS15 strain genome. The authors have included a final summary statement, beginning on line 343, to more clearly state the evidence for the directionality of the events we have described in the manuscript.

- The point of figure 1c is to show that the right part doesn't align well. To be extra careful, it would be best to do a protein alignment, and then back align to get the nucleotides. Likely will give the same result, but then we'll be really confident there's no homology lurking there.

Response

The authors have included an amino acid alignment of Orf9t and Tn916 Orf9 as a fourth panel (figure 1d) to further demonstrate the lack of homology between the two. This new figure is referenced in the text also on line 171. The authors have also reworded the figure legend for figure 1c to clarify the results of the nucleotide sequence alignment between orf9 and orf9t. The lack of homology between the two sequences is explained further in sections 3.3 and 3.4.

Smaller points

- The introduction focuses exclusively on Tn916. I think it would benefit from a little discussion on what is known about the evolution of other CTns. Are there other cases where people have put together scenarios like the one being proposed here?

Response

The authors have included additional information to the introduction, beginning on line 45. This introduction provides further information on how gene acquisition has shaped the evolution of Tn916-like elements as well as other conjugative transposon families.

- Fig 1 b suggestions. Color the E. faecalis Tn916 element differently, and then make the axis below (giving distances in Tn916) have a similar color so that it matches visually.
Also, there is a bit of a gap between the drawing of the genes and this axis, which makes them not group together visually as well as they could.
Make the font of TT31 and E. faecalis Tn916 bigger. This should be prominent.

Response

The authors have amended figure 1b such that the Tn916 diagram has been coloured to follow the  same scheme as the other figures. The gap between the Tn916 diagram and its scale has been reduced and the font of TT313 and E. faecalis Tn916 have been increased.

- Fig 1 c. Since Orf9t was on the top in 1b, it makes sense to put it on the top in 1c.

Response

Figure 1c has been amended such that the orf9t sequence on top of the Tn916 orf9 sequence.

Reviewer 2 Report

The authors describe the identification of a CTn ancestral to Tn916 and how Tn916 and related elements could have been derived from TT31. These are interesting findings on the pool of CTns as well as their evolution. I only have a few minor comments.

- It would be informative to discuss the impact of having both tet(M) and tet(L) in TT31 compared with tet(M) alone in Tn916. In addition, although their putative functions are not known or well-described it would be interesting to add some discussion on the impact of orf9t in TT31 compared with orf 6 orf9 in Tn916.

- It would be interesting to know if the complete TT31 CTn wass reconstructed, it could be found in the current available genomes.

L97-99. Please rephrase as it could be interpreted that xisTn and intTn are not found in Tn916-like elements.

Author Response

Reviewer 2

The authors describe the identification of a CTn ancestral to Tn916 and how Tn916 and related elements could have been derived from TT31. These are interesting findings on the pool of CTns as well as their evolution. I only have a few minor comments.

- It would be informative to discuss the impact of having both tet(M) and tet(L) in TT31 compared with tet(M) alone in Tn916. In addition, although their putative functions are not known or well-described it would be interesting to add some discussion on the impact of orf9t in TT31 compared with orf 6 orf9 in Tn916.

Response

This is a good point and something that we initially speculated on but did not include in our submission. The authors have included a section (4.2 The Impact of tet(L) expression on a Tn916-like element) that discusses the potential impact tet(L) might have on the regulation of excision on such conjugative transposons and the results of having an expanded substrate profile when Tet(L) is present. This section begins on line 240.

- It would be interesting to know if the complete TT31 CTn wass reconstructed, it could be found in the current available genomes.

Response

A reconstructed CTn based on the Tn916-like element identified in the TT31 clone was created by the authors. Both the reconstructed CTn and the original TT31 sequence gave the same search results when they were analysed using BLAST.

- L97-99. Please rephrase as it could be interpreted that xisTn and intTn are not found in Tn916-like elements.

Response

The authors have rephrased this section of the text, on line 120, to clarify that xisTn and intTn are found in Tn916-like elements.